# CRISPR/Cas9 modified *An. gambiae* carrying *kdr* mutation L1014F functionally validate its contribution in insecticide resistance and combined effect with metabolic enzymes

**Linda Grigoraki**[ID]*, **Ruth Cowlishaw**[ID], **Tony Nolan**[ID], **Martin Donnelly**[ID], **Gareth Lycett**[ID], **Hilary Ranson**[ID]*

Vector Biology Department, Liverpool School of Tropical Medicine, Liverpool, United Kingdom

* Linta.Grigoraki@lstmed.ac.uk (LG); Hilary.Ranson@lstmed.ac.uk (HR)

**Data Availability Statement:** All relevant data are within the manuscript and its Supporting Information files.

## Abstract

Insecticide resistance in *Anopheles* mosquitoes is a major obstacle in maintaining the momentum in reducing the malaria burden; mitigating strategies require improved understanding of the underlying mechanisms. Mutations in the target site of insecticides (the voltage gated sodium channel for the most widely used pyrethroid class) and over-expression of detoxification enzymes are commonly reported, but their relative contribution to phenotypic resistance remain poorly understood. Here we present a genome editing pipeline to introduce single nucleotide polymorphisms in *An. gambiae* which we have used to study the effect of the classical kdr mutation L1014F (L995F based on *An. gambiae* numbering), one of the most widely distributed resistance alleles. Introduction of 1014F in an otherwise fully susceptible genetic background increased levels of resistance to all tested pyrethroids and DDT ranging from 9.9-fold for permethrin to >24-fold for DDT. The introduction of the 1014F allele was sufficient to reduce mortality of mosquitoes after exposure to deltamethrin treated bednets, even as the only resistance mechanism present. When 1014F was combined with over-expression of glutathione transferase Gste2, resistance to permethrin increased further demonstrating the critical combined effect between target site resistance and detoxification enzymes *in vivo*. We also show that mosquitoes carrying the 1014F allele in homozygosity showed fitness disadvantages including increased mortality at the larval stage and a reduction in fecundity and adult longevity, which can have consequences for the strength of selection that will apply to this allele in the field.

## Author summary

Escalation of pyrethroid resistance in *Anopheles* mosquitoes threatens to reduce the effectiveness of our most important tools in malaria control. Studying the mechanisms underlying insecticide resistance is critical to design mitigation strategies. Here, using genome modified mosquitoes, we functionally characterize the most prevalent mutation in resistant mosquitoes, showing that it confers substantial levels of resistance to all tested

**Funding:** This study was funded by the Wellcome Trust (https://wellcome.org/), Sir Henry Wellcome Postdoctoral fellowship, Grant reference number: [215894/Z/19/Z] to LG. For the purpose of open access, the author has applied a CC-BY public copyright license to any author accepted manuscript version arising from this submission The funders had no role in study design, data collection and analysis, decision to publish, or preparation of the manuscript.

**Competing interests:** The authors have declared that no competing interests exist.

pyrethroids and undermines the performance of pyrethroid-treated nets. Furthermore, we show that combining this mutation with elevated levels of a detoxification enzyme further increases resistance. The pipeline we have developed provides a robust approach to quantifying the contribution of different combinations of resistance mechanisms to the overall phenotype, providing the missing link between resistance monitoring and predictions of resistance impact.

## Introduction

The widespread use of insecticides in indoor residual spraying and insecticide-treated bednets (ITNs) has been a critical driver in the reduction of malaria cases in the last decades [1]. These tools have been so effective because they reduce the density and the lifespan of mosquitoes and thus their ability to transmit the *Plasmodium* parasite. However, gains in malaria control are now stalling [2] and this has been attributed, at least partially, to increasing levels of insecticide resistance in *Anopheles* vectors [3]. As malaria vector control relies on a limited range of chemicals and new insecticides need years to be developed and approved, it is critical to preserve the effectiveness of available compounds. To do that we need to understand the mechanisms by which insects have evolved resistance and design mitigation strategies.

The molecular basis of insecticide resistance is complex with multiple mechanisms co-existing. The two most widely reported adaptations include mutations at the target site that reduce the insecticide's binding affinity and increased production of detoxification enzymes, like P450s, esterases and GSTs (glutathione transferases) that inactivate the insecticide molecules and enhance their excretion [4]. More recently additional mechanisms have been described, including cuticular modifications that reduce the insecticide's penetration rate [5] and insecticide sequestration from chemosensory proteins [6]. Although in many cases the association of these mechanisms with insecticide resistance is clear, we are still lacking critical knowledge about the effect size of each mechanism and importantly the combined effect of different mechanisms. It has been hypothesized that the synergistic effect of different mechanisms is what leads to operational control failure, [7] but there is little data to support this prediction. This knowledge gap reduces our ability to interpret the results of the various molecular diagnostics that have been developed to screen for the presence of insecticide resistance mechanisms in field populations, simply because the predictive value of the markers used is unclear [8].

Insecticide resistance mechanisms often carry fitness costs. Therefore, removal of the insecticide, as a selection pressure, is predicted to reduce the frequency of resistance alleles in the population. This is the basis of insecticide resistance management strategies that alternate the use of chemistries with different mode of action. However, not all resistance alleles pose high fitness costs, in which case they can persist in populations, and in some cases additional mutations with a compensating role can be selected. Thus, it is important to evaluate the cost of each documented resistance mechanism and make evidence-based decisions on insecticide alternations. A key bottleneck is the ability to compare the effect of specific mutations on defined genetic backgrounds. To date, several studies have documented fitness costs for target-site resistance mutations, but these studies are rarely performed on genetically related strains, which complicates the interpretation of results and often the establishment of a direct link [9].

In this study we examine the contribution to insecticide resistance and the associated fitness costs for mutation L1014F on the *An. gambiae* voltage gated sodium channel (VGSC). Mutation L1014F (or L995F using the *Anopheles gambiae* codon numbering), also known as

classical *kdr* (knock-down resistance), was among the first mechanisms associated with resistance to the organochlorine DDT and to pyrethroids [10]. Pyrethroids are a particularly important insecticide class as they are used in all ITNs, even the newer nets that combine two chemistries, due to their low mammalian toxicity and fast mode of action. Therefore, it is commonplace to genotype for this mutation as an adjunct to resistance prevalence bioassays. Several studies have tried to measure the effect size of L1014F in pyrethroid resistance, either by associating its presence with survival to insecticide exposure [11–13], by introgressing the mutation in an insecticide susceptible strain [14,15] or by genetically modifying *Drosophila melanogaster* [16]. The outcomes of these studies vary, with some reporting a low effect size, while others a moderate to high [17]. However, in all cases the results could be influenced by confounding factors arising either from differences in the genetic background of the mosquito strains compared, or due to the use of a model organism that might not reflect the exact response in mosquitoes.

Here, we have used CRISPR/Cas9 to introduce the L1014F mutation in an *An. gambiae* insecticide susceptible strain providing the opportunity to investigate the direct effect of this mutation on several traits with the minimum possible confounding effects. This is the first time to our knowledge a mosquito strain has been genome edited to functionally validate a single mutation and we provide an experimental pipeline for studies wishing to do the same. We also report the generation of a transgenic *An. gambiae* line in which we have combined the L1014F mutation with over-expression of the detoxification enzyme Gste2, which permits functional validation of the combined effect of target site and detoxification enzymes *in vivo*.

## Results

### Generation of a genome modified *An. gambiae* line bearing the L1014F VGSC mutation in an insecticide susceptible background

Embryos of the insecticide susceptible Kisumu strain were injected with a CRISPR/Donor plasmid mix (Materials and Methods) and 32 larvae ($G_0$) were obtained with RFP (the Red Fluorescent Protein-marker present on the CRISPR plasmid) expression predominantly in their anal papillae, that indicated these fluorescent individuals had actively transcribed plasmids delivered. A subset of each female's $G_1$ progeny was screened for the presence of genetically modified alleles using a LNA diagnostic assay [18]. A positive transformation event was found in progeny of one $G_0$ female at a frequency of 18% (Table A in S1 Text) and verified through sequencing. Sequencing of the *vgsc* region in the transformed line also revealed, based on the presence in the donor construct of silent SNPs (Fig A in S1 Text), that DNA resection during CRISPR reached from one side at least 199bp and from the other side less than 158bp. Thus, even if a gRNA target is not available at or close (within 20 bp) to the mutation site, which is the preferred option, it is still possible to recover transformants with gRNAs targeting a region further away. A summary of the strategy used to generate the Kisumu-F/F line (which is *para*$^{1014F/F}$ homozygous) is illustrated in Fig 1.

### Genome modified mosquitoes carrying the L1014F mutation show increased levels of resistance to all tested pyrethroids and DDT

The extent of the resistance phenotype conferred by mutation L1014F was initially tested using WHO discriminating dose assays. The concentration of insecticides in these assays is fixed at twice the lethal concentration that kills 99% of the susceptible mosquitoes after 1h of exposure and mortality of less than 90% is the threshold to define resistance [19]. Based on these criteria L1014F confers WHO defined resistance to the pyrethroids, permethrin and α -cypermethrin

## A. Embryonic micro-injections and introduction of L1014F mutation through HDR

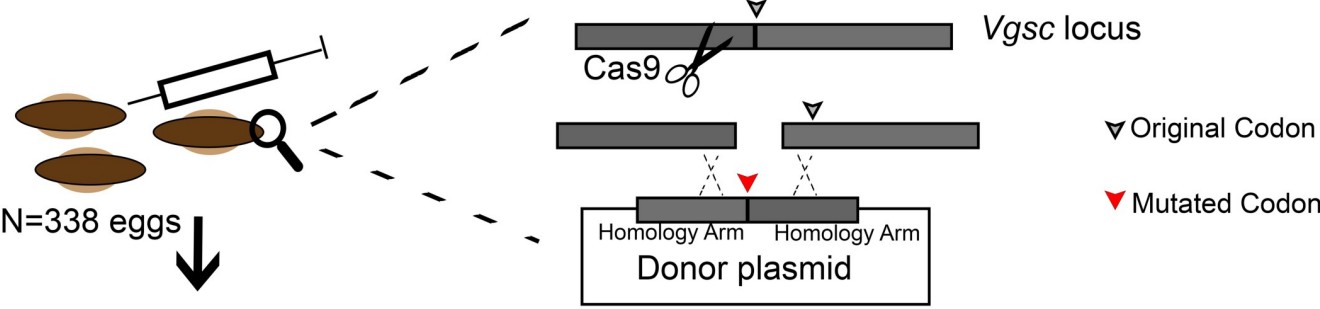

## B. Screening G₀ larvae for transient RFP expression and backcrossing of positives

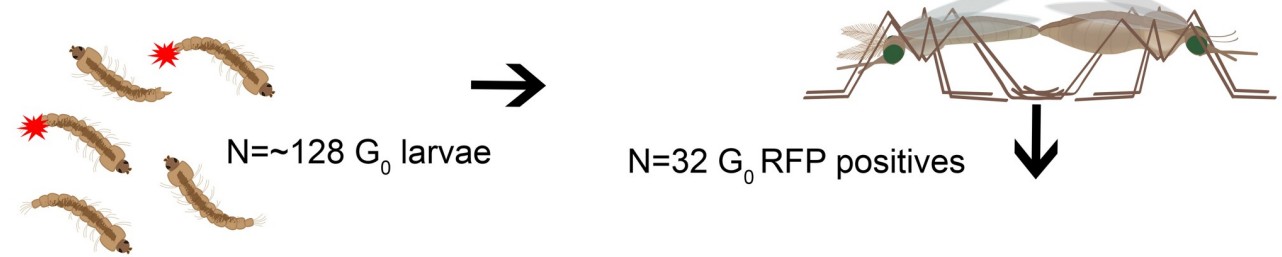

## C. Identification of transformed G₁ progeny using a LNA based diagnostic assay

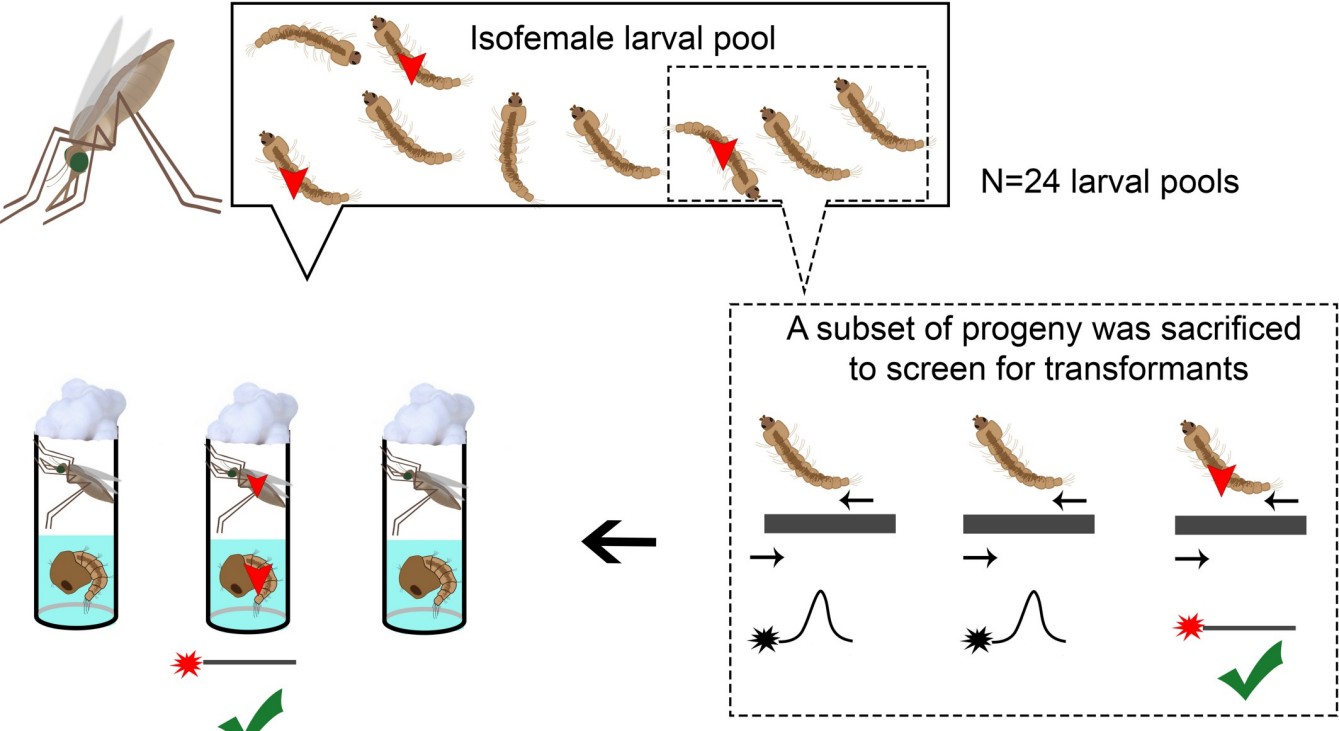

**Fig 1. CRISPR/Cas9 strategy for the generation of Kisumu-F/F line.** A) Injection of Kisumu eggs (In this case N = 338) with an appropriate CRISPR (Cas9 +gRNA)/Donor mix resulted in CRISPR induced homology directed repair (HDR) and conversion of the wild type allele to the mutant L1014F allele. B) Only

individuals with transient expression of RFP in their anal papillae (In this case 32 out of 128 larvae that hatched), which is expressed from the CRISPR plasmid, were pooled in sex-specific founder cages, and backcrossed to Kisumu individuals. C) Females ($G_0$ and Kisumu females mated with $G_0$ males) were left to lay individually (in this case 24 females: 5 $G_0$ females and 19 Kisumu Females laid eggs), then a subset of their progeny (max 30%) was sacrificed and screened in pools of 2 larvae each with a LNA assay for identification of positive transformants ($G_1$). A single positive $G_1$ pool was identified. The remaining siblings in the pool were reared until the pupal stage and transferred to individual tubes to emerge. Post-eclosion the pupal cases were used to screen for the presence of the mutant allele. Individuals that carried the mutant allele were used to establish the line.

only in homozygosity (recessive character) (Fig 2B and 2D), while it confers resistance to DDT in heterozygosity (dominant character) (Fig 2A). A reduction in mortality for Kisumu-F/F individuals against deltamethrin, was observed although not statistically significant (Fig 2C). The percentage of mosquitoes being knocked down immediately after the 1h exposure period was also recorded for each insecticide (S1 Data and Fig B in S1 Text). In all cases except delta-methrin, there is significant inhibition of knock down after 1hr exposure in F/F mosquitoes, but in F/L genotypes a knockdown resistance phenotype was only observed against DDT. There was no significant difference between 1hr knock down and 24 hr mortality for any

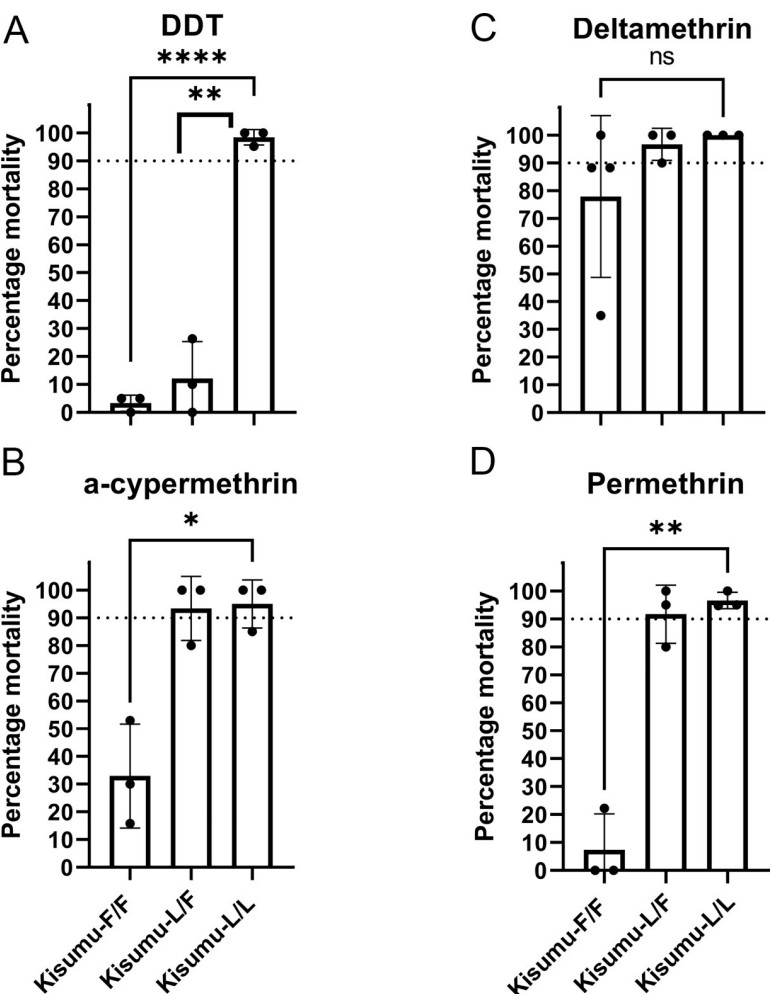

**Fig 2. Measurement of sensitivity to insecticides using WHO tube bioassays.** 2–5-day old females were exposed to standard WHO discriminating doses for 1h and mortality recorded 24h later. Error bars represent the SD (At least 3 replicates of 20 mosquitoes each were used per strain). The dotted line marks the WHO 90% mortality threshold for defining resistance. Welch's t test with P < 0.05 significance cutoff. **P < 0.01; ***P < 0.001; ****P < 0.0001.

**Table 1.** *Time response bioassay results.*

| Insecticide | Strain | $LT_{50}$ (St. Error) | Upper and Lower Limits | $RR_{50}$ |
|---|---|---|---|---|
| *Deltamethrin* | Kisumu-F/F | 31.5min (3.3) | 38.2–24.8 | 14.6 (19.3–10.0) |
| | Kisumu-L/F | 9.1 min (1.4) | 12.0–6.2 | 4.2 (5.9–2.5) |
| | Kisumu-L/L | 2.1 min (0.25) | 2.6–1.6 | |
| *Permethrin* | Kisumu-F/F | 160.5 (11.0) | 182.5–138.4 | 9.9 (13.2–6.6) |
| | Kisumu-L/F | 24.3 (2.5) | 29.5–19.1 | 1.5 (2.0–0.95) |
| | Kisumu-L/L | 16 (2.4) | 20.9–11.2 | |
| *a-cypermethrin* | Kisumu-F/F | 77.5 (3.7) | 84.9–70.1 | 19.7 (23.6–15.8) |
| | Kisumu-L/F | 12.9 (1.2) | 15.4–10.3 | 3.2 (4.1–2.4) |
| | Kisumu-L/L | 3.9 (0.33) | 4.5–3.2 | |
| *DDT* | Kisumu-F/F | >540min | | >24.5 |
| | Kisumu-L/F | ~211.6 (10.8) | 189.8–233.3 | 9.6 (8.2–10.9) |
| | Kisumu-L/L | 22.0 (1.3) | 24.6–19.4 | |

The $LT_{50}$ (time required to obtain 50% mortality) values are given for each strain. Resistant Ratios ($LT_{50}$ resistant strain/ $LT_{50}$ control strain) are given in comparison to Kisumu-L/L. Upper and Lower limits represent the 95% fiducial limits of the $LT_{50}$. For each time point at least 3 replicates of 20 female mosquitoes 2–5 day old were used per strain. Raw data provided in S1 Data.

insecticide or genotype (S1 Data and Fig B in S1 Text). As expected, no resistance was observed against the carbamate insecticide bendiocarb (S1 Data) that targets the acetylcholinesterase and not the VGSC.

To further quantify the level of resistance conferred by L1014F to the insecticides we performed time response assays and estimated the time required to obtain 50% mortality ($LT_{50}$) for each of the three genotypes (Table 1). Kisumu-F/F exhibited the highest predicted Resistance Ratio ($RR_{50}$) against DDT (> 24.5-fold) although accurate determination of the $LT_{50}$ for DDT was not possible in mutant homozygotes, as mortality of Kisumu-F/F was less than 23% after 9 hours of exposure to DDT, which was the latest time point measured. The $RR_{50}$ for α-cypermethrin was 19.7-fold, for deltamethrin 14.6-fold and for permethrin 9.9-fold. Lower $RR_{50}$ were observed for the heterozygote (Kisumu-L/F) mosquitoes in all cases (Table 1), thus L1014F under the exposure conditions used has a semidominant effect on resistance.

## L1014F reduces mortality of mosquitoes after contact with deltamethrin treated ITNs

To assess the impact of the L1014F mutation on ITN effectiveness, we performed standard WHO cone bioassays exposing females of the Kisumu-F/F strain and the Kisumu-L/L strain to PermaNet 2.0 (impregnated with deltamethrin) and control untreated nets, for 3min. 100% mortality was recorded for the Kisumu-L/L females 24h after exposure to PermaNet2.0, while the mean mortality rate for the Kisumu-F/F strain was significantly less at ≈65% (*P = 0.0003*, Welch's test)(Fig 3). No mortality was recorded in the control untreated nets.

## Pre-exposure to PBO increases mortality of the Kisumu-F/F strain to deltamethrin

Piperonyl butoxide (PBO) significantly enhances the efficacy of pyrethroids by inhibiting the function of P450 detoxification enzymes and/or by acting as an adjuvant, enhancing insecticide uptake [20]. Here we used the Kisumu-F/F strain to demonstrate that PBO pre-exposure significantly reduces deltamethrin resistance, even in the absence of metabolic resistance. In the absence of PBO, 25min exposure to the standard WHO discriminating dose of

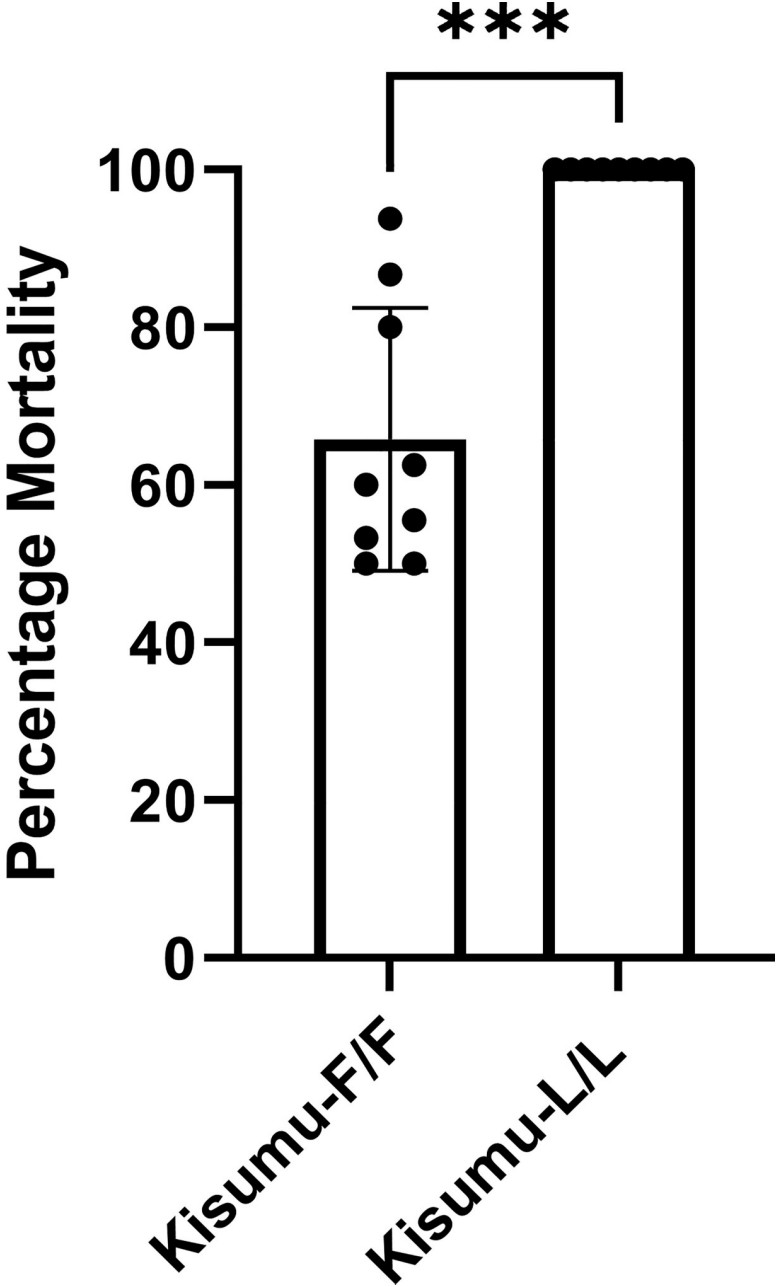

**Fig 3. Cone Bioassay with PermaNet 2.0.** The percentage mortality 24h after exposure is depicted for the Kisumu-F/F and Kisumu strains. Nine replicates of five females each were tested. Error bars show SD. Welch's t test with P < 0.05 significance cutoff. **P < 0.01; ***P < 0.001.

deltamethrin resulted in a mean mortality of 13.7±7.9%, while 1h pre-exposure to PBO resulted in a significant increase in mortality (mean mortality of 84.7±0.86%) (Fig 4).

## The L1014F mutation carries fitness costs

We investigated the effect of mutation L1014F on several life history traits. In terms of growth, a significantly lower percentage of Kisumu-F/F larvae (mean of 75%) reached the pupae stage

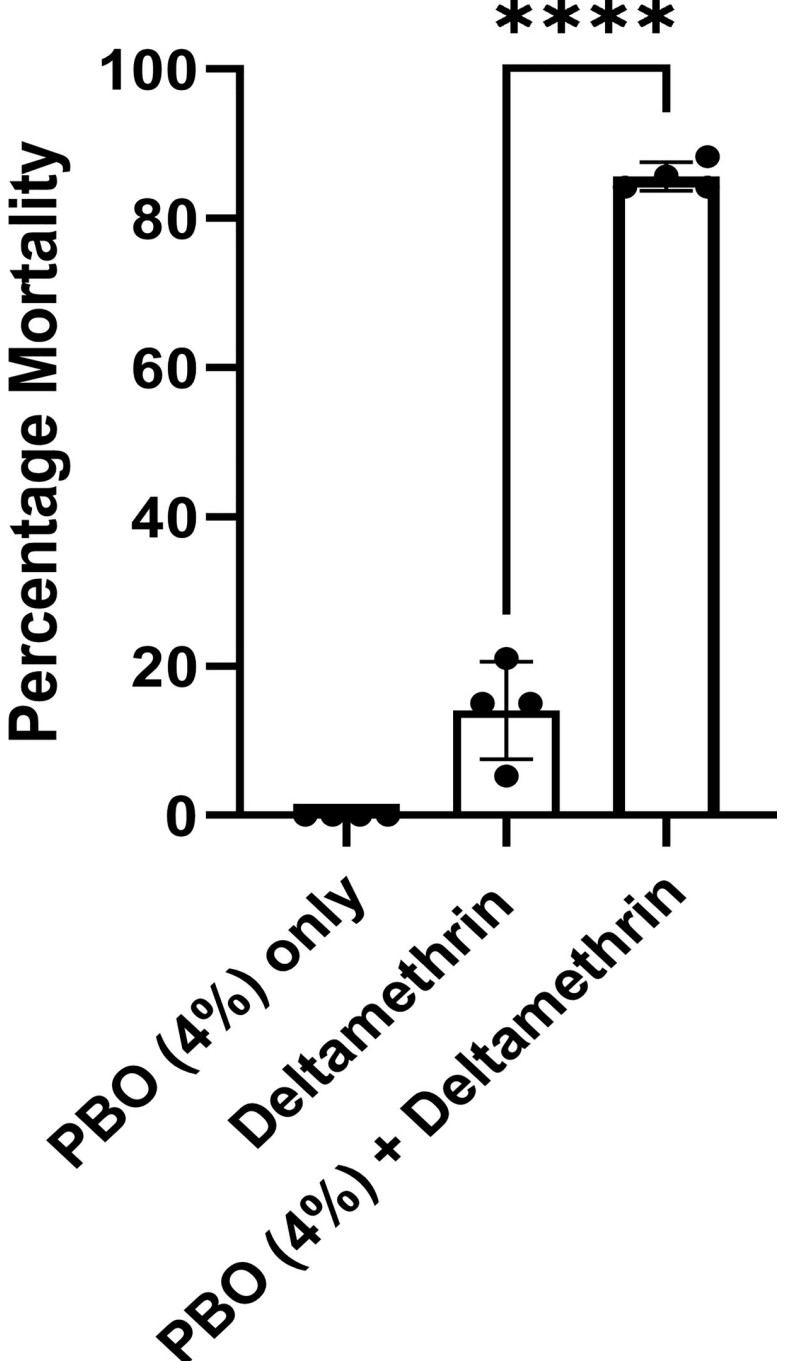

**Fig 4. Mortality to Deltamethrin with and without pre-exposure to PBO.** The percentage mortality is shown 24h after: A) 1h exposure to 4% PBO, B) 25min exposure to 0.05% deltamethrin and C) 1h exposure to 4% PBO immediately followed by 25min exposure to 0.05% deltamethrin. Four replicates of 15–21 females, 2–5 days old each were used. Error bars show SD. Welch's t test with $P < 0.05$ significance cutoff. ** $P < 0.01$; *** $P < 0.001$; **** $P < 0.0001$.

compared to wild type Kisumu-L/L (mean of 91%; *P<0.0001*) (Fig 5A). In addition, the fecundity of Kisumu-F/F females was significantly reduced with 35.7% of females not producing eggs, after one blood meal, compared to 10.7% for the Kisumu-L/L females (*P = 0.003*) (Fig

5B). In contrast the fertility (number of eggs laid) was not significantly different between the two genotypes (mean number of eggs laid by Kisumu-F/F was 72 (±24 SD) compared to 83 (± 37 SD) for Kisumu-L/L) (Fig 5C), nor the number of larvae that hatched (mean number of larvae hatched for each Kisumu-F/F female 54 (±26 SD) compared to 55 (±30 SD) for Kisumu-L/ L (*P >0.05*) (Fig 5D). Finally, the lifespan of female Kisumu-F/F was significantly reduced (median age decreased from 24 to 21 days) compared to Kisumu-L/L females (*P = 0.02*) (Fig 5E).

## Over-expression of Gste2 increases resistance of mosquitoes carrying mutation L1014F

To investigate the combined effect of target site mutations and detoxification enzymes we used a previously generated [21] transgenic strain ubiquitously over-expressing the detoxification enzyme Gste2, and through genetic crosses with the Kisumu-F/F strain generated a line: (Kisumu-F/F)/Gste2 ($para^{1014F/F}$; Ubi-A10GAL4:UAS-GSTe2) carrying both resistance mechanisms in an otherwise susceptible genetic background. The mean mortality of individuals from the (Kisumu-F/F)/Gste2 line was 17.2% after 160min exposure to permethrin treated WHO papers, compared to a mean mortality of 65.5% for the Kisumu-F/F line and 100% for the Gste2 overexpressing line (Fig 6), showing the combined effect of these two mechanisms.

## Discussion

*In vivo* functional validation is critical in establishing the importance of candidate mechanisms, alone and in combination, in phenotypic insecticide resistance. Here we have used CRISPR/Cas9 to introduce the VGSC mutation L1014F, which is widespread in insecticide resistant *An. gambiae* populations in Africa, in an insecticide susceptible genetic background. Ordinarily genome modification strategies use a dominant fluorescent marker to identify rare transformants. However, studying the role of single nucleotide polymorphisms precludes the introduction of additional sequences, as these could confound the interpretation of results. In this study we show that the CRISPR/Cas9 strategy we followed achieves transformation rates high enough to recover transformants without the aid of a dominant marker gene and thus can be used in *An. gambiae* to study the role of single nucleotide polymorphisms. Its applicability to other mosquito species will depend on the rates of homology directed repair, which can be lower than we achieved in *An. gambiae* [22]. The use of CRISPR/Cas9 to study the effect of target site mutations in insecticide resistance has been predominantly performed using the model organism *Drosophila melanogaster* [16, 23–25] and in one agricultural pest [26], thus our study reports, to the best of our knowledge, the first time this approach being used in a major insect vector of human disease pathogens. Several target-site resistance mutations have been described in these insects whose effect size needs to be clarified, but which could be assessed following the approach described here; these include mutations in acetylcholinesterase, the target of organophosphate and carbamate insecticides, the GABA-gated chloride channel that is associated with dieldrin resistance [27] and the chitin synthase that is associated with resistance to diflubenzuron and other benzoylurea insecticides inhibiting this enzyme [25,28].

Introduction of L1014F in the Kisumu susceptible background increased insecticide resistance against all pyrethroid insecticides tested and DDT, with resistance levels ranging from 9.9-fold for permethrin to >24.5-fold for DDT in quantitative assays. The very high levels of resistance to DDT, observed even for heterozygous individuals, suggests that L1014F was originally selected by the widespread use of DDT and retained in the populations after introduction of pyrethroids, by conferring lower, but still substantial levels of resistance. Permethrin and

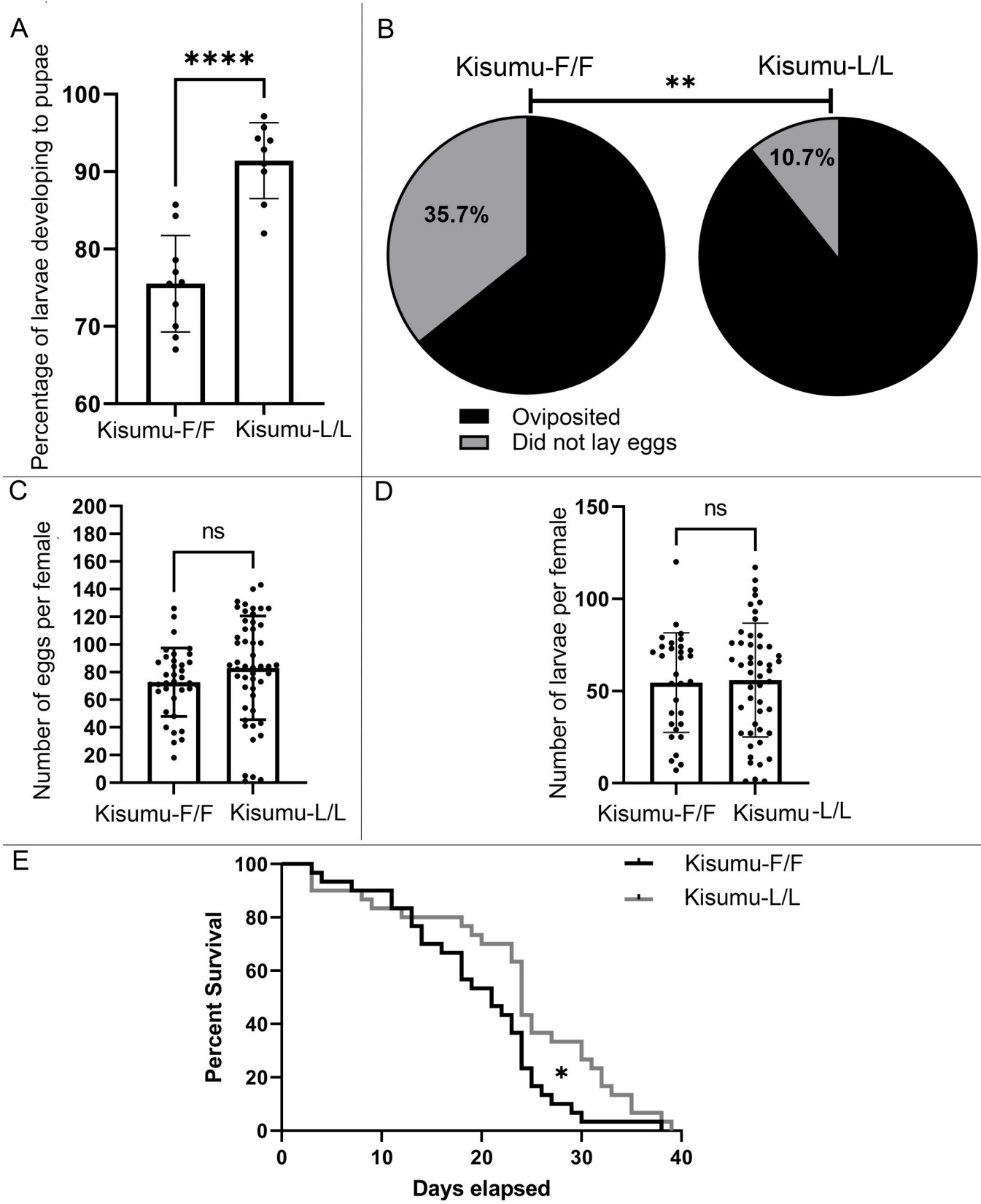

**Fig 5. Evaluating the pleiotropic effects of L1014F on different life history traits.** A) Difference in the mean percentage of L1 larvae reaching the pupae stage is shown for the Kisumu-F/F (75%) and Kisumu-L/L (91%) strains. 10 replicates were tested in total, 7 replicates of 70 larvae each and 3 replicates of 200 larvae each. Error bars represent the SD. Unpaired t-test, P <0.0001. B) Pie charts showing the difference in fecundity (percentage of females that oviposited at least one egg vs females that did not). The total number of females tested per strain was 56. Fisher's exact test, P = 0.0031. C)The mean number of eggs laid per female was not statistically significant between the Kisumu-F/F (N = 72) and Kisumu (N = 83) strains. Error bars represent the SD. Mann-Whitney (P = 0.062). D) the mean number of larvae that hatched per female is shown for the two strains. No significant difference was observed. Mann-Whitney-test (P = 0.91). E) Longevity of female mosquitoes under standard lab conditions. Thirty females in total were tested per strain. Mantel-Cox test (P = 0.02),*P $\leq$ 0.05; **P $\leq$ 0.01; ***P $\leq$ 0.001; ****P $\leq$ 0.0001.

deltamethrin resistance observed in Kisumu-F/F is very similar to the levels of insensitivity L1014F confers to *Ae. aegypti* VGSCs expressed in the Xenopus oocyte system (8-fold for permethrin and 14-fold for deltamethrin) [29]. In that system insensitivity is calculated based on the percentage of channels whose activity is being modified by pyrethroids. In the case of deltamethrin although the quantitative bioassay showed a 14-fold increase in resistance, similar to the 13-fold reported for transgenic *Drosophila* carrying the equivalent mutation [16], the WHO bioassay involving 1h exposure to a discriminating dose did not define Kisumu-F/F mosquitoes as resistant (mortality <90%). Thus, the WHO discriminating dose for deltamethrin seems to have a higher threshold than the doses recommended for other pyrethroids, making it difficult to compare resistance between compounds using the standard 1h time point. Moreover, the WHO diagnostic assay failed to detect resistance in heterozygotes to any of the pyrethroids, whereas heterozygous mutants were clearly resistant to DDT with this test. Quantitative bioassays however showed resistance of heterozygotes for all tested pyrethroids ranging from 1.5-fold for permethrin to 4.2-fold for deltamethrin. This is a clear illustration of the limitation that standard WHO bioassays face in identifying populations where 1014F is increasing in frequency, if only pyrethroid sensitivity is assayed, given that even a population with 100% heterozygotes could be classified as susceptible to the three main pyrethroid insecticides. Our WHO and quantitative bioassay data also demonstrate that characterizing resistance as a recessive or dominant trait depends on the type of assay and conditions used. For example, the 1014F allele is dominant for DDT resistance using the standard WHO bioassay, but recessive based on the quantitative analysis.

We also showed that the 1014F allele when in homozygosity is sufficient to induce reduced mortality after mosquitoes are exposed to commercial, deltamethrin treated ITNs. We recorded a 67.5% mortality in the standard WHO cone bioassay for Kisumu-F/F. This is higher than the 26% mortality that was previously reported [14] for the kdr-Kisumu line, which was generated by introgressing the L1014F mutation from a resistant field strain into Kisumu [15]. Although introgression substantially dilutes the resistance genetic background, it is impossible, even with multiple rounds of crossing, to achieve the resolution that allows one to look at the effects of a mutation in isolation. This is particularly relevant for centromeric loci located in low recombination regions, such as the *vgsc* [30]. In addition, whole genome sequencing data from the *Anopheles gambiae* 1000 Genomes Project, have shown a high genetic variation in the *vgsc* gene itself. Twenty non-synonymous substitutions were identified, thirteen of which were found to occur almost exclusively on haplotypes carrying the L1014F resistance allele and may enhance or compensate for the L1014F resistance phenotype [31]. Thus, the higher levels of resistance obtained for the introgressed line could be related to the presence of additional unscreened mutations in the *vgsc* allele originating from the parental resistant population or the presence of other genes that are carried over during introgression.

The P450 inhibitor PBO is commonly used to estimate the contribution of P450 metabolism to resistance phenotypes. Here we showed that pre-exposure to PBO increased mortality of the Kis-L1014F strain after deltamethrin exposure. This enhancement in deltamethrin efficacy, even in the absence of metabolic resistance, could be either mediated through the

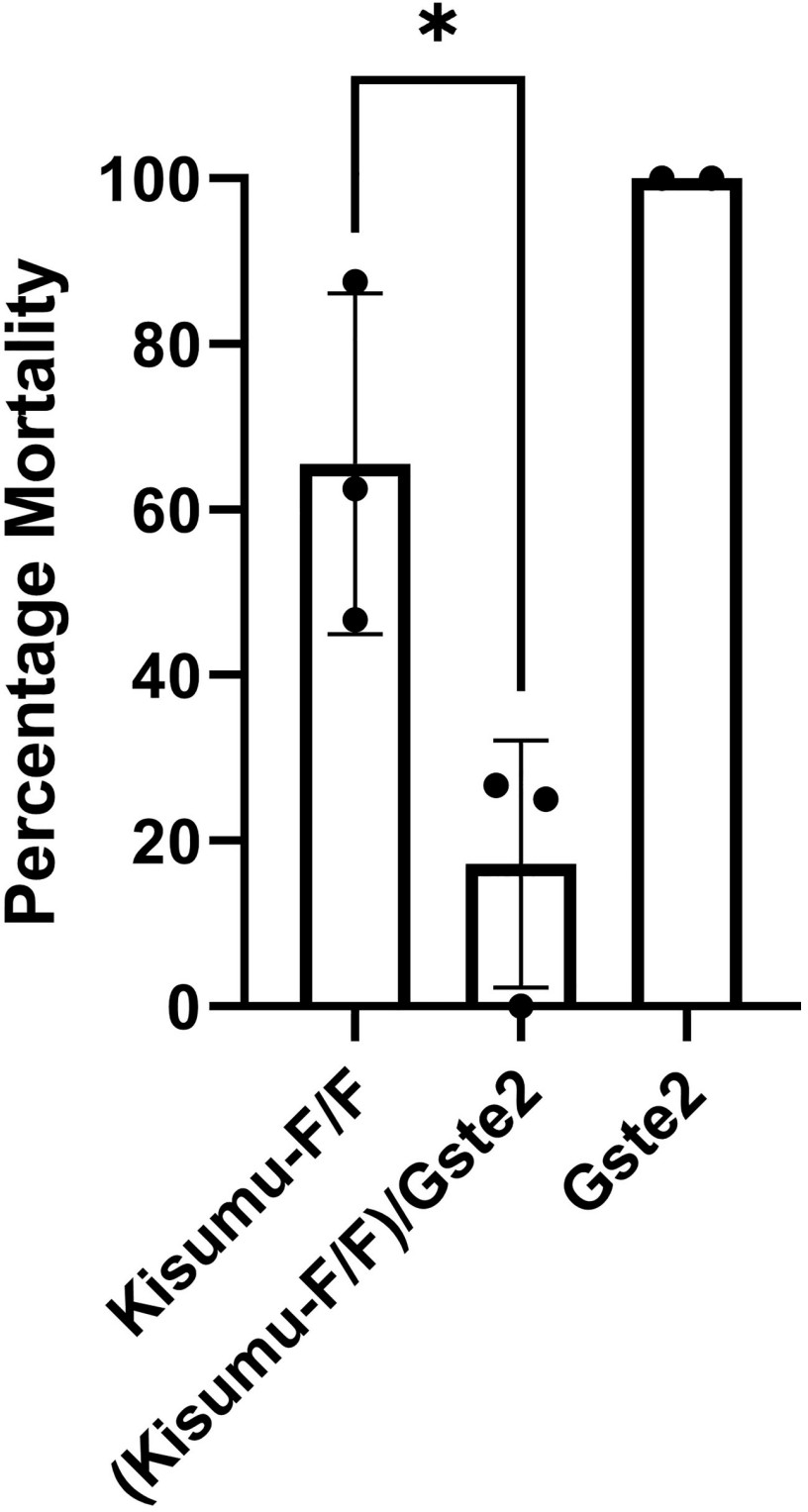

**Fig 6. Multi-tissue over-expression of Gste2 further reduces mortality of mosquitoes from the Kisumu-F/F strain to permethrin.** WHO tube bioassay representing mortality rate after 160min of exposure. The Gste2 line (Ubi-A10GAL4:UAS-GSTe2) showed close to 100% mortality after 1h exposure, as shown by Adolfi et al [21] and 100% mortality after 160min. Three replicates of 15–20 female mosquitoes each were used for the Kisumu-F/F and (Kisumu-F/F)/Gste2 lines and two replicates of 15 female mosquitoes for the Gste2 line. Error bars represent SD. *Welch's t test with P = 0.03.*

inhibition of the endogenous levels of P450s, that might still contribute to resistance when in combination with the L1014F mutation, or by enhancing the penetration, and thus bioavailability, of the insecticide through the cuticle [20]. Either way, this result suggests that demonstration of elevated P450 levels is not a necessary pre-requisite for use of PBO containing products to mitigate against resistance, as has previously been recommended.

Combining mutation L1014F with over-expression of the detoxification enzyme Gste2 resulted in increased levels of resistance to permethrin. Transgenic mosquitoes over-expressing Gste2 (allele carrying the I114T variant associated with DDT resistance [32]), as the only resistance mechanism have previously been shown to be resistant to DDT and the organophosphate insecticide fenitrothion, but not to permethrin or deltamethrin [21]. However metabolic assays with recombinant *An. funestus* Gste2 have shown its ability to directly metabolise DDT and permethrin *in vitro* [33]. Thus, the *An. gambiae* Gste2 is either a poor permethrin metaboliser that needs the presence of target-site resistance to show a phenotypic effect or it mainly metabolises secondary metabolites of permethrin that can also have toxic effects. We had intended to test the combined effect of Gste2 and L1014F in DDT resistance, but the insensitivity of the Kisumu-F/F line even after long exposure times and high doses of DDT in topical applications, precluded detection of any further increase in resistance with the addition of Gste2. However, the permethrin data support the hypothesis [16] that the different mechanisms can interact in *An. gambiae*, and that their combined effect ultimately shapes the resistance phenotype in mosquitoes. Although the mechanisms of this interaction have not been elucidated yet, there are several hypotheses as described in [16]. The kdr mutation, by reducing the binding affinity of the insecticide, could provide more time for detoxification enzymes to act and avoid saturation. Alternatively, the detoxification enzymes could reduce the number of parental insecticide molecules reaching the nervous system or generate less-toxic metabolites with an even further reduced binding affinity to the mutated VGSC.

In addition to the impact on insecticide resistance we showed that L1014F has pleiotropic effects on fitness related traits. It has long been hypothesized that L1014F carries fitness costs [34], but providing a direct association had been complicated by the comparison of populations with different genetic backgrounds [35]. Although we cannot rule out the possibility of CRIPSR induced off-targets or haplotype-specific effects in the Kisumu-F/F line, the comparison we perform involves the least possible confounding genetic effects. We show that significantly fewer individuals from the Kisumu-F/F strain are able to develop from early instar larvae to pupae. This could also explain, at least partially, the lower than expected number of homozygous individuals observed in the process of establishing the Kisumu-F/F line. In addition, a significantly lower number of females from the Kisumu-F/F line oviposited. Although we did not measure insemination rates, this may be related to the reported association of VGSC mutations with reduced male mating success [36], and warrants further investigation in future. A reduced lifespan was also observed for Kisumu-F/F females, which could be even more pronounced under field conditions. However, it should be noted that fitness costs related to the L1014F mutation could be ameliorated by compensatory mechanisms in wild populations. For example, several additional non-synonymous mutations are found on the *vgsc* gene in linkage with L1014F, which is consistent with them either conferring additional resistance

or compensating for the deleterious effects of L1014F [31,37, 38], and can be examined by further mutagenesis of the Kisumu-F/F line generated herein.

## Conclusions

Understanding the mechanisms of insecticide resistance and their phenotypic contribution can provide predictive value for molecular diagnostic markers to be used as resistance prevalence monitoring tools and assist in resistance control program decision making. This work reports the functional validation of mutation L1014F, which is widely used as a marker of pyrethroid resistance in malaria vectors, but without a clear understanding of how precisely it affects the performance of insecticide-based vector control tools. We show that L1014F even in isolation, provides substantial levels of insecticide resistance that impact the performance of ITNs, but critically resistance levels are further increased when combined with the overexpression of a metabolic enzyme, the Gste2. This highlights the importance of interpreting molecular diagnostics carefully, considering the synergistic or additive effects of the different mechanisms. Further work is needed to investigate the complex interactions of the different insecticide resistance mechanisms and the different mutations on the *vgsc* gene; the ability, demonstrated here, to introduce precise and defined mutations in any combination, on any given haplotype offers a powerful tool to finally dissect these interactions. This in turn will greatly increase the predictive value of multi-locus diagnostic panels to detect and track the emergence of insecticide resistance, with the potential to be transformative for insecticide resistance management programs.

## Materials and methods

### Mosquito rearing

All mosquitoes were maintained under standard insectary conditions, at 26˚C ± 2˚C and 70% relative humidity ± 10% under L12:D12 hour light:dark photoperiod. All stages of larvae were fed on ground fish food (Tetramin tropical flakes, Tetra, Blacksburg, VA, USA) and adults were provided with 10% sucrose solution *ad libitum*.

### Genome modification strategy

CRISPR/Cas9 coupled with homology directed repair (HDR) was used to introduce the L1014F VGSC mutation in the insecticide susceptible *An. gambiae* strain, Kisumu.

### Generation of Cas9 plasmid

The p174 plasmid generated in [39], carrying: a human-codon-optimized Cas9 under the control of the germline specific *zpg* (zero population growth) promoter, a 3xP3::RFP marker and a U6::gRNA spacer cloning cassette was used. p174 was digested with BsaI and specific gRNA spacers (listed in Table B in S1 Text) targeting the *vgsc* locus near the L1014 codon, were introduced under the control of the U6 polIII promoter. This was done through Golden Gate cloning of annealed oligos with compatible overhangs (Table B in S1 Text). Identification of multiple gRNA target sites and assessment of off-targets was performed using the ZiFiT (http://zifit.partners.org/) and ChopChop (https://chopchop.rc.fas.harvard.edu) websites. As described in Fig A in S1 Text, the guide RNA that proved successful in generating mutants was gRNA 2.

## Generation of Donor for HDR

A 1,600 bp region of the Voltage Gated Sodium Channel (*vgsc*, AGAP004707-RA) gene having the 1014 codon in the middle (homology arms extending 800 bp either direction) was synthesized *de novo* by Genescript in a puc19 vector, including the A→T transversion that generates the L1014F mutation (codon alteration TTA→TTT) and other synonymous SNPs in the gRNA target sites to avoid cleavage of the vector by Cas9 (Fig A in S1 Text).

## Mosquito embryo injections and identification of $G_1$ transformants

Freshly laid eggs of the Kisumu strain were microinjected with a mix of the Cas9 and donor plasmids (300ng/ul each). In order to enrich for those individuals injected with appreciable quantities of the Cas9 and donor plasmid mix, surviving $G_0$ larvae expressing RFP (present on the Cas9 plasmid as a 3xP3:RFP) transiently in the anal papillae were reared separately and backcrossed with the Kisumu strain in sex-specific cages. Females of both crosses were put in individually egg-laying tubes and a subset of their progeny ($G_1$) was used to screen for positive transformants. DNA was extracted from $G_1$ larvae in pools of two by grinding them in 30ul of STE buffer (0.1M NaCl, 10mM Tris-HCl pH 8, 1mM EDTA pH 8), followed by a 25min incubation at 95˚C. 2ul of DNA extract was used in a previously established LNA based diagnostic assay for the L1014F mutation [18].

## Establishment of Kisumu-F/F line

Where genome-modified alleles were identified in progeny of a $G_1$ female, siblings were left to develop and added in individual emergence tubes once they reached the pupae stage. The day after adult emergence the pupae case left behind was lysed in 10 ul of STE buffer at 95 ᵒC for 25min. 2ul were used to identify heterozygotes with the LNA based diagnostic assay. Heterozygotes were backcrossed once more with the Kisumu strain in sex-specific cages to obtain higher numbers of individuals carrying the mutant allele ($G_2$). Heterozygote $G_2$ individuals were intercrossed to obtain homozygote $G_3$ individuals. A sufficiently high number of homozygotes able to establish the strain were obtained after several generations of intercrossing, as the number of homozygous mutants was lower than expected, possibly due to the observed fitness costs. The locus of the genome edited strain was verified by sequencing the PCR product of the L1014externalF1 and L1014externalR1 primers (Table B in S1 Text) at Genwiz.

## Establishment of (Kisumu-F/F)/ Gste2 line

Males of the Gste2 line generated and described in Adolfi et al.,2019 [21], expressing the Gste2 metabolic enzyme (allele including the I114T substitution, associated with DDT resistance [32]) under an endogenous polyubiquitin promoter:Gal4 fusion (A10) and marked with 3xP3 driven yellow fluorescent protein, were crossed with females of the Kisumu-F/F line. G1 male progeny, heterozygotes for Ubi-A10GAL4:UAS-Gste2 and L1014F were crossed with females of the Kisumu-F/F strain. Once progeny of this cross reached the pupae stage, they were first screened under a fluorescent microscope for a YFP signal. YFP positive individuals were left to emerge in individual tubes. The pupae case left behind was used to extract DNA and screen through the L1014F LNA assay, as described in the previous paragraph, for homozygotes. Individuals from the (Kisumu-F/F)/Gste2 line used in toxicity bioassays were 1014F/F, but a mix of heterozygotes (one copy) and homozygotes (two copies) for Ubi-A10GAL4:UAS-Gste2.

## Assessment of insecticide susceptibility

**WHO tube bioassays.**   WHO tube bioassays were performed as described in [40] using insecticide impregnated papers of standard discriminating doses: 0.75% permethrin, 0.05% deltamethrin, 0.1% bendiocarb, 4% DDT and 4% PBO (obtained from Universiti Sains Malaysia). To determine the $LT_{50}$ (exposure time resulting in 50% mortality) we varied the exposure time (S1 Data). Knock down was scored immediately after exposure and mortality after a 24hour recovery time. At least 3 replicates of 20 females (2–5 days old) each were performed for each time point. A control tube with no insecticide was included in each test for each strain. $LT_{50}$, Resistance Ratio and associated statistical parameters were calculated from log-logistic 2 parameter dose response models using the drc package from [41] and the drm(), compParm(), ED() and EDcomp() functions in R statistical software (version 3.4.3).

**Cone bioassays.**   Cone bioassays were performed as described in [42] using PermaNet 2.0 nets (obtained directly by Vestergaard). Nine replicates of five female mosquitoes, 2–5 days old were tested using at least three randomly selected pieces of the nets (control and treated). Mosquitoes were exposed for 3min and then transferred to recovery cups. Mortality was recorded 24h later.

## Life history traits

**Development to pupae.**   Seven replicate trays (23cm x 34cm x 7cm) containing 70 larvae each and three replicate trays containing 200 larvae each (for testing possible differences in pupation rate depending on larvae densities) were set up for the Kisumu and Kisumu-F/F strains. Larvae of the two strains were treated in the same way, fed equal amount of TetraMin fish food (using standard size spatulas) and reared under standard insectary conditions. The number of individuals reaching the pupae stage was recorded every day. A two-tailed unpaired student's t-test was used (GraphPad Prism 9.0.0) to compare pupation rates.

**Fecundity and fertility.**   Three replicate mating buckets each containing 20 females and 30 males were set up for the Kisumu and Kisumu-F/F homozygous strains. Mosquitoes were fed five days later with our standard 'blood' supply (50,50 mix of human research red cells and plasma, NHS blood and transplant service) using a Hemotek membrane feeding system. Those that did not feed were removed from the experiment. Two days post blood feeding females were placed in individual laying cups containing 20 ml of water. Females were aspirated out of the cups two days later and oviposited eggs were counted under a stereoscope. The number of females that laid (at least one egg) and did not lay were recorded and analyzed using a Fisher's exact test (GraphPad Prism 9.0.0). Larvae that hatched were counted four days later to include any with a delayed hatching. Statistical comparison of egg and larvae counts was done using a Mann-Whitney U test (GraphPad Prism 9.0.0).

**Adult longevity.**   Thirty females from each strain, were transferred in pools of five in six replicate cups and fed *ad libitum* with 10% (w/v) sugar solution. Mortality was recorded every day and dead mosquitoes removed. Longevity was assessed using GraphPad Prism 9.0.0 using a Mantel-Cox test.

## Supporting information

**S1 Text. CRISPR/Cas9 strategy for generating Kisumu-F/F. Fig A**: Strategy for the construction of the CRISPR/Cas9 donor plasmid. The nucleotide sequence of the central part (600bp) of the *An. gambiae vgsc* (Gene ID: AGAP004707) fragment (1,600bp in total) used to construct the CRISPR/Cas9 donor plasmid for HDR is depicted. Light orange areas correspond to exons 18 and 19. Position 2,422,652 (based on the AgamP3 reference sequence, chromosome arm 2L) where an A>T transversion generates the TTA—>TTT codon alteration creating

mutation L1014F is marked with an asterisk. Underlined sequences correspond to the selected CRRISPR/Cas9 targets (gRNAs 1,2,3 and 4). Green areas mark the PAM (-NGG) triplets for each gRNA in the target sequence. Red letters show the SNPs introduced in the donor sequence. These include silent mutations in the gRNA target site to avoid cleavage of the donor plasmid from Cas9 and the A>T transversion creating mutation L1014F. Letters below red letters show the original sequence. Sequencing of the region in individuals from the Kisumu-F/F strain revealed that silent mutations introduced in the target sequences of gRNA's 2,3 and 4 were retained while those of gRNA 1 were not. **Fig B**: Comparison of knock-down and mortality. The percentage of mosquitoes being knocked-down (immobile or unable to stand or take off) immediately after exposure to standard WHO assays (1h exposure time) was recorded, as well as the mortality 24h later based on standard WHO criteria (2). Error bars represent the SD (At least 3 replicates of 20, 2–5 day old female mosquitoes each were used per strain). **Table A**: Summary of screening $G_1$ progeny to identify positive transformants. **Table B**: Primers used.
(DOCX)

**S1 Data. Raw data for toxicity bioassays and fitness cost experiments.**
(XLSX)

## Acknowledgments

We would like to thank Beth Poulton (LSTM) for assistance with the R code used to analyze the time response assays and her help in rearing the Gste2 transgenic colony. We would also like to thank Fraser Colman (LSTM) for long term maintenance and provision of the Gste2 transgenic line, Sara Elg (LSTM) for assistance with maintenance of the Kisumu lines and Manuela Bernardi (LSTM) for providing the mosquito illustrations used in Fig 1.

## Author Contributions

**Conceptualization:** Linda Grigoraki, Martin Donnelly, Hilary Ranson.

**Formal analysis:** Linda Grigoraki.

**Funding acquisition:** Linda Grigoraki.

**Investigation:** Linda Grigoraki, Ruth Cowlishaw.

**Methodology:** Linda Grigoraki, Ruth Cowlishaw, Gareth Lycett.

**Resources:** Tony Nolan, Gareth Lycett.

**Writing – original draft:** Linda Grigoraki, Hilary Ranson.

**Writing – review & editing:** Ruth Cowlishaw, Tony Nolan, Martin Donnelly, Gareth Lycett.

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
