## [Decision Letter · Decision Letter 0]

19 May 2021

Dear Dr Grigoraki,

Thank you very much for submitting your Research Article entitled 'CRISPR/Cas9 modified An. gambiae carrying kdr mutation L1014F functionally validate its contribution in insecticide resistance and interaction with metabolic enzymes.' to PLOS Genetics.

The manuscript was fully evaluated at the editorial level and by independent peer reviewers. The reviewers appreciated the attention to an important problem, but raised some concerns about the data presented in Figures 2&3. Based on the reviews, we will not be able to accept this version of the manuscript, but we would be willing to review a revised version. We cannot, of course, promise publication at that time.

If you decide to revise the manuscript for further consideration at PLOS Genetics, please aim to resubmit within the next 60 days, unless it will take extra time to address the concerns of the reviewers, in which case we would appreciate an expected resubmission date by email to plosgenetics@plos.org.

[LINK]

We are sorry that we cannot be more positive about your manuscript at this stage. Please do not hesitate to contact us if you have any concerns or questions.

Yours sincerely,

Subba Reddy Palli, Ph.D.

Associate Editor

PLOS Genetics

Gregory P. Copenhaver

Editor-in-Chief

PLOS Genetics

Reviewer's Responses to Questions

**Comments to the Authors:**

Reviewer #1: Acceptable for publication. Would have been nice to see LD50 data for the insecticides.

Reviewer #2: This manuscript describes the use of genome editing to introduce the pyrethroid resistance mutation L1014F into the VGSC of Anopheles gambiae. The impact of this on insecticide sensitivity and organismal fitness was assessed. Furthermore the interaction of this mutation with a metabolic mechanism, over-expression of glutathione transferase Gste2, was investigated.

The manuscript is very well written and the study innovative and important. The kdr mutation has been linked to resistance in An gambiae for more than two decades, however, the precise level of resistance conferred by this mechanism, in isolation or combination with other mechanisms, has remained unclear. This study emphatically demonstrates its importance in conferring resistance while also uncovering fitness costs associated with the mutations presence.

The results are thoroughly and carefully discussed and any potential limitations/confounding effects (i.e. from off-target effects of CRISPR-CAS) noted.

In summary I recommend this paper is accepted and the following typos corrected in production.

Line 192: change: Mutation L1014F carries fitness costs. To ‘The L1014F mutation carries fitness costs’.

Line 261: correct the grammer in the following sentence: The resistance observed for Kisumu-F/F against permethrin and deltamethrin are very similar to the levels of insensitivity mutation L1014F provides to Ae. aegypti VGSCs expressed in the in vitro Xenopus oocyte system (8-fold for permethrin and 14-fold for deltamethrin) (29).

Reviewer #3: In this paper, Grigoraki et al. generate an Anopheles gambiae transgenic line bearing a kdr mutation to further validate that kdr contributes to insecticide resistance. Using CRISPR/cas9 system, they edit the genome of a susceptible mosquito strain to introduce single nucleotide polymorphism (kdr mutation L1014F). They further test the mortality of this line using different insecticide compounds and confirm kdr mutation L1014F increases resistance to insecticides. Overexpression of gste2, a detoxification enzyme, in the kdr mutant line increase the resistance phenotype. Finally, they analyse the effect of the mutation on mosquito fitness. This paper is well written, clear and conclusions supported by the data. However, I do not think that this study reveals novel methods nor biological insights as it only confirms previous studies showing kdr is involved in resistance to insecticides and therefore, may be better suited for a more specific journal.

Minor comments

Why is there a lot of variation in F/F mosquitoes mortality after ITN exposure (Fig 3) with some mosquito groups showing a mortality above 90%, ie are susceptible?

The resistance to deltamethrin varies a lot as well (eg fig2 with 70-80% mortality vs fig4 with less than 20%). How can this be?

**Have all data underlying the figures and results presented in the manuscript been provided?**

Reviewer #1: Yes

Reviewer #2: Yes

Reviewer #3: Yes

PLOS authors have the option to publish the peer review history of their article (what does this mean?). If published, this will include your full peer review and any attached files.

Reviewer #1: No

Reviewer #2: No

Reviewer #3: No

---

## [Editor Report · Decision Letter 1]

1 Jun 2021

Dear Dr Grigoraki,

Thank you very much for submitting your Research Article entitled 'CRISPR/Cas9 modified An. gambiae carrying kdr mutation L1014F functionally validate its contribution in insecticide resistance and combined effect with metabolic enzymes.' to PLOS Genetics.

The manuscript was fully evaluated at the editorial level and the editors ask that you address the comment from Reviewer 1 of the previous version and include the LD50 data for insecticides.

[LINK]

Yours sincerely,

Subba Reddy Palli, Ph.D.

Associate Editor

PLOS Genetics

Gregory P. Copenhaver

Editor-in-Chief

PLOS Genetics

---

## [Editor Report · Decision Letter 2]

8 Jun 2021

Dear Dr Grigoraki,

We are pleased to inform you that your manuscript entitled "CRISPR/Cas9 modified An. gambiae carrying kdr mutation L1014F functionally validate its contribution in insecticide resistance and combined effect with metabolic enzymes." has been editorially accepted for publication in PLOS Genetics. Congratulations!

Yours sincerely,

Subba Reddy Palli, Ph.D.

Associate Editor

PLOS Genetics

Gregory P. Copenhaver

Editor-in-Chief

PLOS Genetics

Comments from the reviewers (if applicable):

**Data Deposition**

http://datadryad.org/submit?journalID=pgenetics&manu=PGENETICS-D-21-00525R2

**Press Queries**

---

## [Editor Report · Acceptance letter]

29 Jun 2021

PGENETICS-D-21-00525R2 

CRISPR/Cas9 modified *An. gambiae* carrying *kdr* mutation L1014F functionally validate its contribution in insecticide resistance and combined effect with metabolic enzymes. 

Dear Dr Grigoraki, 

We are pleased to inform you that your manuscript entitled "CRISPR/Cas9 modified *An. gambiae* carrying *kdr* mutation L1014F functionally validate its contribution in insecticide resistance and combined effect with metabolic enzymes." has been formally accepted for publication in PLOS Genetics! Your manuscript is now with our production department and you will be notified of the publication date in due course.

With kind regards,

Katalin Szabo

PLOS Genetics

On behalf of:
